# Circuit-Based Design of Microfluidic Drop Networks

**DOI:** 10.3390/mi13071124

**Published:** 2022-07-16

**Authors:** Nassim Rousset, Christian Lohasz, Julia Alicia Boos, Patrick M. Misun, Fernando Cardes, Andreas Hierlemann

**Affiliations:** Bioengineering Laboratory, Department of Biosystems Science and Engineering, ETH Zürich, CH-4058 Basel, Switzerland; christian.lohasz@bsse.ethz.ch (C.L.); julia.boos@bsse.ethz.ch (J.A.B.); patrick.misun@bsse.ethz.ch (P.M.M.); fernando.cardes@bsse.ethz.ch (F.C.); andreas.hierlemann@bsse.ethz.ch (A.H.)

**Keywords:** hanging-drop network, standing-drop network, capillary pressure, hydrostatic pressure, hydraulic-circuit analogy, fluid shear stress

## Abstract

Microfluidic-drop networks consist of several stable drops—interconnected through microfluidic channels—in which organ models can be cultured long-term. Drop networks feature a versatile configuration and an air–liquid interface (ALI). This ALI provides ample oxygenation, rapid liquid turnover, passive degassing, and liquid-phase stability through capillary pressure. Mathematical modeling, e.g., by using computational fluid dynamics (CFD), is a powerful tool to design drop-based microfluidic devices and to optimize their operation. Although CFD is the most rigorous technique to model flow, it falls short in terms of computational efficiency. Alternatively, the hydraulic–electric analogy is an efficient “first-pass” method to explore the design and operation parameter space of microfluidic-drop networks. However, there are no direct electric analogs to a drop, due to the nonlinear nature of the capillary pressure of the ALI. Here, we present a circuit-based model of hanging- and standing-drop compartments. We show a phase diagram describing the nonlinearity of the capillary pressure of a hanging drop. This diagram explains how to experimentally ensure drop stability. We present a methodology to find flow rates and pressures within drop networks. Finally, we review several applications, where the method, outlined in this paper, was instrumental in optimizing design and operation.

## 1. Introduction

Recent advances in microfabrication techniques, such as the 3D printing of new types of resins, have significantly increased the design potential for microfluidic devices, allowing for almost arbitrarily complex interweaving channel networks [1,2]. Optimizing the design can be performed by modeling the behavior of fluids within microfluidic channel networks with intricate numerical methods, such as the finite element method. However, owing to computational limits, 3D models of large microfluidic networks cannot be handled efficiently, particularly when exploring the effects of several design and operation parameters. Therefore, a separate “first-pass” method is required for predicting flow within such networks before establishing more intricate 3D mathematical models.

A powerful method, used for such initial device optimization, is the hydraulic-circuit analogy [3]. This analogy has found various applications in the field of microfluidics [4,5,6]. This approach is based on working with volumetric flow rates Q and fluid pressures p as if they were electric currents and voltages, respectively. While this approach is commonly used for microfluidic channels with completely closed channels, it is rarely used for open fluidic devices with dynamic drop volumes. One of the reasons is that a drop that changes in size cannot be modeled with simple electrical components, such as constant resistors or sources, but require variable components to represent the nonlinear dependency between drop geometry and capillary and hydrostatic pressures.

In this review article, we show the methodology, developed in our group to apply the hydraulic-circuit analogy for the purpose of designing and operating microfluidic hanging- and standing-drop networks.

First, we describe the basic useful microfluidic components and all relevant variables to model them. Then, we discuss the practical limitations of drop networks, with a focus on hanging-drop networks. We then lay out a strategy to draw a microfluidic-network scheme that helps in defining the set of equations describing the microfluidic circuit. Solving the set of equations then provides a description of drop volume dynamics and helps to find critical drop behaviors that need to be considered when designing and operating drop networks. Finally, we show how this method can be and has been applied in the context of different and previously published open drop-based microfluidic networks.

## 2. Basic Microfluidic-Drop-Network Components

We first present the main elements used for designing and drawing microfluidic networks: hydraulic resistances, fluidic sources, atmospheric pressure, and drop compartments.

### 2.1. Hydraulic Resistance

Hydraulic resistance is directly analogous to resistive elements within electric circuit design. Solving the Navier–Stokes equation for flow through a microfluidic channel results in the velocity vector field v(x,y,z) and the pressure scalar field p(x,y,z). The flow rate Q through the microfluidic channel is found by integrating the velocity vector field over the channel cross-sectional area Ω (Figure 1a). This cross section is typically set at the inlet or outlet of the channel, where the flow should be orthogonal to Ω ($n_{\Omega }$), which then yields a proportional relationship between Q and the pressure drop between the inlet and outlet of the channel $\Delta p=p_{in}-p_{out}$.

This proportional relationship is called the hydraulic resistance of the channel R and is entirely dependent on the channel design and on the viscosity η of the liquid phase flowing through the channel. The most general definition of this relationship is given in [3] with the perimeter P and the area A of the cross section Ω, and L the length between the inlet and outlet of the channel.
(1)$$Q\equiv \overset{}{\underset{\Omega }{\int }}v\left(x,y,z\right)\cdot n_{\Omega }d\Omega \to \Delta p\approx \left(2\eta L\frac{P^{2}}{A^{3}}\right)Q\to \Delta p=RQ$$

We include useful definitions of hydraulic resistances for circular $R_{O}$, high-aspect-ratio rectangular $R_{∥}$, and rectangular $R_{◼}$ channel cross sections that remain constant through the channel length (Figure 1b).
(2)$$R_{O}=\frac{8}{\pi }\frac{\eta L}{a^{4}}$$
(3)$$R_{∥}=\frac{12\eta L}{h^{3}w}$$
(4)$$R_{◼}=\frac{12\eta L}{h^{3}w}{\left[1-\overset{\infty }{\underset{n,odd}{\sum }}\frac{1}{n^{5}}\frac{192}{\pi^{5}}\frac{h}{w}\tanh\left(n\pi \frac{w}{2h}\right)\right]}^{-1}\approx \frac{12\eta L}{h^{3}w}\frac{1}{1-0.63\frac{h}{w}}$$

These are the basic geometries that are typically used in the literature to design microfluidic devices, where the width w is defined to be larger than the height h of a channel.

### 2.2. Fluidic Source

A fluidic source either acts as a fluid volume source or drain, as is the case in the flow-through operation of microfluidic chips, or acts as an internal actuator of flow or pressure, as is the case in recirculating operation of microfluidic chips. Each fluidic source uses a force to generate a pressure that drives flow (Table 1). Examples are: (i) surface tension causing a capillary pressure $p_{c}$ that generates a pressure differential between two material phases, e.g., at an air–liquid interface (ALI), or (ii) gravity causing a hydrostatic pressure $p_{g}$ that generates a pressure differential between two elements at different heights h. A summary of common fluidic sources, the forces driving fluid motion in these sources, their definitions, and the physical limitations on the achievable flow rates is provided in Table 1 and schematically illustrated in Figure 2a–c.

At their core, all fluidic sources are pressure-driven. “True” pressure-driven sources apply a constant pressure that is a function of their geometry. They are limited by the design properties of the source itself. For example, capillary-driven flow is limited by the minimum achievable fluid surface curvature (Figure 2c), and gravity-driven flow is limited by the maximum height difference generating pressure differentials (Figure 2a). With pressure-driven flow, pressure differentials through the liquid phase of various elements drive fluid motion.

Conversely, with flow-driven flow, liquid motion generates pressure differentials through the liquid phase. Therefore, flow-driven sources apply a variable pressure in order to ensure a constant flow rate. They are limited by the mechanical properties of the chambers generating the flow. For example, peristaltic pumps are limited by the compliance of the peristaltic tubing, leading to unpredictable flow rates for highly resistive microfluidic devices. Syringe pumps are limited by the strength of the syringe itself, which can shatter, or the water tightness of the plunger driving fluid motion in the syringe, which can leak.

### 2.3. Atmospheric Pressure

For the purpose of microfluidic-network design, atmospheric pressure takes the role of the “ground” in the electric circuit design. Atmospheric pressure acts as a reference pressure and ensures that we can accurately compute the absolute pressure in each microfluidic element. Regardless of the driving force of fluid motion, pressure differentials play a crucial role in predicting microfluidic network behavior. However, the absolute value of the pressure within different microfluidic elements can cause various unwanted effects. In closed microfluidic devices, high pressure can cause failure of microfluidic channels, akin to Blaise Pascal’s barrel bursting due to hydrostatic pressure [7]. In open microfluidic devices, high pressure can cause a seemingly static system to burst and leak in a highly dynamic manner (Appendix A).

### 2.4. Drop Compartments

A circular opening in an open microfluidic network is referred to as a “drop compartment”. Drop compartments can either be “hanging” (Figure 3a) or “standing” (Figure 3b), depending on the chip concept and the experimental configuration. A drop compartment of radius a can be accurately modeled in a microfluidic circuit as a capillary pressure source $p_{c}$, linked to the network $p_{n}$ through a hydrostatic pressure source $p_{g}$ and a resistance $R_{c}$ (Figure 3c,d). We approximate the resistance $R_{c}$ as that of a circular channel of a length equal to the drop height h.
(5)$$p_{c}=\frac{2\gamma }{r},\;p_{g}=\rho gh,\;R_{c}\approx \frac{8}{\pi }\frac{\eta h}{a^{4}}$$

The singular difference between a standing and hanging drop, for the purpose of a hydraulic circuit design, is the sign of the hydrostatic pressure. Flow toward the ALI $Q_{c}$ is therefore driven by $\Delta p=p_{n}+p_{g}-p_{c}$ for a hanging drop, and $\Delta p=p_{n}-p_{g}-p_{c}$ for a standing drop (Figure 3c,d).

In practice, since $R_{c}$ is low relative to the channel hydraulic resistance, the air–liquid drop interface acts as a compliant element. Drops react almost instantly to gradual changes in fluid pressure due to hydrostatic and hydrodynamic forces. However, drop behavior is particularly difficult to predict when submitted to rapid pressure changes, e.g., when tilting a chip, elevating an inlet reservoir, or upon step changes in flow rates, due to the higher channel resistance in the microfluidic-drop network. The drop behavior upon sudden pressure changes needs to be recapitulated with the modeling methodology outlined in this section. An exercise to intuit drop dynamics is to follow a step increase in the drop compartment pressure $p_{n}$ in a standing drop:A standing-drop compartment pressure increase drives flow toward the ALI $Q_{c}$;○$Q_{c}>0$ increases the drop volume V → ΔV>0;○ΔV>0 reduces the ALI radius (equation r(V,a) in Table 2);○ΔV>0 increases the drop height (equation h(V,a) in Table 2);
Drop geometry changes (r and h as highlighted in Figure 3a,b) induce pressure changes:○A reduction in the ALI radius increases the capillary pressure $p_{c}$;○An increase in the relative height increases the hydrostatic pressure $p_{g}$;An increase in $p_{c}$ and $p_{g}$ changes the pressure differential $\Delta p=p_{n}-p_{g}-p_{c}$;A reduction in pressure differential reduces the flow rate toward the ALI.

This process repeats iteratively until the pressure differential is null and can be described by the following differential equation:(6)$$\frac{dV}{dt}=\frac{p_{n}-p_{g}-p_{c}}{R_{c}}=\frac{p_{n}}{R_{c}}-\frac{\rho gh\left(V,a\right)}{R_{c}}-\frac{1}{R_{c}}\frac{2\gamma }{r\left(V,a\right)}$$

Looking at the expression for r(V,a) and h(V,a) in the full-page Table 2, Equation (6) is difficult to solve analytically, even for a single drop. The highly nonlinear relation between drop volume and pressure makes it impossible to model the drop using only constant electrical components. Instead, hydrostatic and capillary pressures need to be modeled using variable pressure sources, which iteratively change according to differential Equation (6). Therefore, for a more complex problem, e.g., a series of drops in a microfluidic network, we used numerical methods.

## 3. Practical Limitation of Drop Systems

The main limitation of drop systems is when the drop becomes too large, causing leakage from the microfluidic network. For the sake of simplification, we chose a hanging-drop setup to explain this phenomenon, referred to herein as “drop crash”. In this case, drop crash, explained with Figure 4, is when sustained hydraulic $p_{n}$ and hydrostatic $p_{g}$ pressure on the ALI exceed the capillary pressure $p_{c}$ (Figure 3a), causing the hanging drop to fall off the chip. However, a standing-drop setup also suffers from this limitation; albeit, $p_{g}$ works with $p_{c}$ in keeping drop integrity (Figure 3b). Excessive flow rates, however, can easily generate a $p_{n}$ that exceeds $p_{c}+p_{g}$.

Generally, if a pressure increase causes a flow that increases the drop volume past a hemisphere, then the ALI radius starts to increase. Therefore, instead of the capillary pressure increasing to balance internal hydraulic pressure, it is reduced and diverges from equilibrium. This nonlinear divergence drives flow into the drop at an increasing rate. The divergence can then cause a drop to grow past the limit, given by the critical capillary length $l_{c}=\sqrt{\gamma /\rho g}$ [8], which is characteristic of a fluid of a given density ρ subjected to a gravitational acceleration g. We call this phenomenon a “crash” of the system, as gravitational forces overtake capillary forces on the drop ALI.

Figure 4 explains the dynamics of drop crashing. Figure 4a shows, on the y-axis, the pressure, normalized by the maximum capillary pressure of a hanging drop, versus, on the x-axis, the volume within the hanging drop, normalized by the volume of a hemispherical drop. The normalization and equations giving the plotted curves are presented in the Appendix A. The black line, showing the normalized drop capillary pressure $\tilde{p_{c}}$, is drawn with Appendix A. The dashed black line, showing normalized internal pressure $\tilde{p_{n}+p_{g}}$, is drawn with Appendix A.

An exercise to interpret Figure 4a is to follow the ${\tilde{p}}_{c}$ starting at a $\tilde{V}=0.02$, i.e., an empty drop (Figure 4d). In this case, the $\tilde{V}$ increases stably until $p_{c}=p_{n}+p_{g}$. Significantly increasing Δh from Figure 4d leads to the state plotted in Figure 4e. In this case, $\tilde{V}$ increases slowly until $\tilde{V}=1$ (minimal difference between $p_{c}$ and $p_{n}+p_{g}$); then, $\tilde{V}$ rapidly increases until the drop crashes (divergent difference between $p_{c}$ and $p_{n}+p_{g}$). Reducing Δh from Figure 4d leads to the state plotted in Figure 4f. In this case, $\tilde{V}$ decreases slowly until $p_{c}=p_{n}+p_{g}$.

An unstable volume increase (Figure 4e) can be recuperated by reducing the drop height to a technical minimum of Δh=0. However, if this is performed too late, as would be the case with a large extrahemispherical drop, it leads to the state plotted in Figure 4g. In this case, a volume decrease is impossible, and the drop is irrecuperable. A pseudostable drop condition exists, but any perturbation will cause a drop crash.

In practice, Figure 5 can be consulted, where the state of a drop is entirely defined by experimental pressure and volume. Figure 5a shows a linear representation of the normalized pressure. Acting experimentally on a drop network changes the pressure $p_{n}$ resulting in varied internal pressures (Figure 5a dashed lines). We plot $p_{n}$ on the y-axis of Figure 5b. The observable dimension of a drop is its volume $\tilde{V}$ on the x-axis of Figure 5. The response of a drop is seen as a color axis with the pressure difference $\Delta p=p_{n}+p_{g}-p_{c}$. The drop dynamics is either a volume increase (red) or decrease (blue) until it reaches the stable black curve. This results in the phase diagram for a drop compartment of a given diameter (2a = 3 mm). The critical drop volume above which gravitational forces cause a drop crash is also shown with a dashed red line.

The drop dynamics explained in this section are crucial for developing an intuition for manipulating drop systems. We include other phase diagrams for several drop compartment apertures in Appendix A.

## 4. Methodology for Modeling Drop Networks

We will describe the method that we elaborated to model drop platforms with the basic elements of microfluidic-drop networks that we defined and detail their limitations. We first define the known and unknown variables of the model that we then use to define the microfluidic network scheme. This microfluidic network scheme allows us to define the set of equations describing the microfluidic circuit. Solving the set of equations with a matrix approach allows us to gain valuable insights into microfluidic-drop-network design and operation. The described methodology is generally applicable to any microfluidic-drop network, but we will demonstrate it on a specific example, referred to as the “scalable microphysiological system”.

### 4.1. Variable Definition

Known and unknown variables must be properly defined and enumerated to define the set of equations that describe fluid flow within the microfluidic network and to numerically solve the unknown variables. All variables are summarized in Table 3.

#### 4.1.1. Known Variables

Known variables are either design- or operation-related. They include hydraulic resistances, prescribed flow rates, capillary pressures, and hydrostatic pressures.

**Hydraulic resistance** is defined by the channel design. Microfluidic channels typically have a constant resistance given by Equations (2), (3), or (4), depending on their cross section and channel length. However, reservoirs [9], drops [10,11,12,13], and other elements that can fill or empty over time will lead to varying channel lengths and, therefore, are modeled as continuously variable hydraulic resistances. The filling and emptying dynamics of such reservoirs is discussed in Appendix A. Additionally, valving mechanisms [14] will induce a step decrease or increase in the resistance of a channel, either allowing or restricting flow along a specific path.

**Flow rates** can be prescribed during the operation of a microfluidic device. Mechanical fluidic sources (Table 1) are the only sources able to supply a steady flow rate. The flow rate can be constant when a continuous flow through the microfluidic network is defined, e.g., for continuous sampling [10,15], substance dosing [16], or medium replenishment [11]. The prescribed flow rate can also vary, e.g., for applying varying shear stresses [17] or periodically increasing liquid turnover [11].

**Capillary pressures** are an operational variable, given by the geometry of a curved ALI, which are determined by liquid cohesion and interfacial adhesion forces. Capillary pressure varies throughout an experiment to comply with liquid pressure within a drop compartment. However, if a steady state of flow is reached, e.g., in flow-through applications [11], then the capillary pressure can be constant.

**Hydrostatic pressure** is an operational variable that is given by the relative vertical position of a microfluidic element. Hydrostatic pressure can be constant throughout an experiment, where, if the chip is kept static, the level of the chip returns a hydrostatic pressure map through the device, which induces a change in volume from drop to drop. Hydrostatic pressure can also vary, as is the case with gravity-driven flow, e.g., upon tilting the chip, upon unequally filling the chip’s inlet and outlet reservoirs, or upon connecting an inlet to an elevated reservoir [9].

#### 4.1.2. Unknown Variables

Unknown variables need to be defined and computed to recapitulate microfluidic network dynamics. They include flow rates through each microfluidic element and pressures at each node between microfluidic elements.

**Flow rates** need to be defined between each node of the microfluidic circuit. The direction of the flow and the corresponding flow rate must be set, where a negative flow rate indicates a reversed flow. The best strategy is to define flow rate variables according to each hydraulic resistance, mirroring the resistance nomenclature.

**Pressures** need to be defined at each node and are used to compute flow rates through the device. Pressure decreases linearly in the direction of the flow across each microfluidic channel with a constant cross section.

### 4.2. Microfluidic-Network Scheme

Knowing the common microfluidic elements, defined in Table 3, allows for drawing the electric equivalent circuit of a microfluidic-channel network. We show an example of a circuit design for a published standing-drop chip (Figure 6a) [9]. The standing-drop chip consists of 10 standing-drop compartments (Figure 6b), arranged in series and flanked by reservoirs on both sides (Figure 6a). The standing drops have a defined geometry (Figure 6c) allowing the definition of flow resistances through the chip. A single-compartment circuit is shown (Figure 6d) and repeated N=10 times. The full circuit is schematically shown (Figure 6e), where each open surface is connected to atmospheric pressure. Figure 6e is used to define the set of circuit equations. The strategy to draw the electric equivalent circuit is elaborated in the Appendix A.

### 4.3. Circuit Equations

As soon as a circuit for a microfluidic network is fully defined, the corresponding set of equations can be written down in a matrix form. Equations take two forms: (i) equations describing the flow through resistive elements, linking pressure drops to flow rates, and (ii) equations describing mass conservation at nodes.

#### 4.3.1. Resistive Pressure Drop

The relationship between hydraulic resistance, flow rate, and pressure must be satisfied in every part of the microfluidic network that features flow. This results in a number of equations that is equal to the number of resistive elements in the device. Equations of the resistive pressure drop are of the form $p_{i}-p_{j}=Q_{ij}R_{ij}$, where the pressure drop from node i to node j drives a flow rate $Q_{ij}$ through the resistive element $R_{ij}$.

#### 4.3.2. Fluidic Nodal Rule

The nodal rule, also known as Kirchhoff’s current law, ensures conservation of charge within a circuit. Equivalently, the fluidic nodal rule ensures the conservation of volume within a microfluidic network. This rule looks at each node and equates all influxes in that node to its outfluxes. This results in a number of equations that is equal to the number of nodes in the device. Equations of the fluidic nodal rule are of the form $Q_{hi}=Q_{ci}+Q_{ij}$, where the flow rate $Q_{hi}$ flowing from node h to node i equals the sum of the flow rate $Q_{ci}$ flowing from node i to drop i and the flow rate $Q_{ij}$ going from node i to node j (Figure 6d, blue arrows).

#### 4.3.3. Circuit Matrix

The resistive pressure drop and fluidic nodal rule are used to define a set of equations. The number of defined equations should be equal to the number of unknown variables. If there are more equations, then the system is overdetermined, which happens when equations are linearly dependent. Linearly dependent equations should be identified and combined. The set of equations are then written in a matrix notation Ax=y. The known vector y contains the known pressures or flow rates. The unknown vector x contains all unknown flow rates and all unknown pressures. The characteristic matrix A typically contains hydraulic resistances $R_{i}$ in most of the diagonal elements and −1 or 1 in the relevant off-diagonal elements. Here, we show the resulting matrix equation Ax=y for the example of a gravity-driven tilting chip, shown in Figure 6, with N=3.
(7)$$\left[\begin{array}{cccccccccc} R_{in} & & & & & & & 1 & & \\ & R_{12} & & & & & & -1 & 1 & \\ & & R_{23} & & & & & & -1 & 1\\ & & & R_{out} & & & & & & -1\\ & & & & R_{c1} & & & -1 & & \\ & & & & & R_{c2} & & & -1 & \\ & & & & & & R_{c3} & & & -1\\ 1 & -1 & & & -1 & & & & & \\ & 1 & -1 & & & -1 & & & & \\ & & 1 & -1 & & & -1 & & & \end{array}\right]\left[\begin{array}{c} Q_{in}\\ Q_{12}\\ Q_{23}\\ Q_{out}\\ Q_{c1}\\ Q_{c2}\\ Q_{c3}\\ p_{1}\\ p_{2}\\ p_{3} \end{array}\right]=\left[\begin{array}{c} p_{g-in}\\ 0\\ 0\\ -p_{g-out}\\ -p_{c1}-p_{g1}\\ -p_{c2}-p_{g2}\\ -p_{c3}-p_{g3}\\ 0\\ 0\\ 0 \end{array}\right]$$

The set of equations describing a pressure-driven microfluidic-drop network typically yields a symmetric sparse matrix. Rigorously defining the circuit, variables, and equations, which describe the flow within a microfluidic network, gives significant freedom in redesigning the channels, reconfiguring the network, and changing the device operation conditions. For example, the chip from Figure 6, described by Equation (7), can be operated with a continuous flow rate instead of being gravity-driven [16]. In this case, the known variable $p_{g-in}$ from Equation (7) becomes an unknown inlet pressure $p_{in}$, and the unknown variable $Q_{in}$ from Equation (7) becomes a known inlet flow rate $Q_{in}$. The resulting sparse matrix is no longer symmetric:(8)$$\left[\begin{array}{cccccccccc} -1 & & & & & & & 1 & & \\ & R_{12} & & & & & & -1 & 1 & \\ & & R_{23} & & & & & & -1 & 1\\ & & & R_{out} & & & & & & -1\\ & & & & R_{c1} & & & -1 & & \\ & & & & & R_{c2} & & & -1 & \\ & & & & & & R_{c3} & & & -1\\ & -1 & & & -1 & & & & & \\ & 1 & -1 & & & -1 & & & & \\ & & 1 & -1 & & & -1 & & & \end{array}\right]\left[\begin{array}{c} p_{in}\\ Q_{12}\\ Q_{23}\\ Q_{out}\\ Q_{c1}\\ Q_{c2}\\ Q_{c3}\\ p_{1}\\ p_{2}\\ p_{3} \end{array}\right]=\left[\begin{array}{c} -Q_{in}R_{in}\\ 0\\ 0\\ -p_{g-out}\\ -p_{c1}-p_{g1}\\ -p_{c2}-p_{g2}\\ -p_{c3}-p_{g3}\\ -Q_{in}\\ 0\\ 0 \end{array}\right]$$

Reconfiguring the matrix and solving it with the new inlet resistances $R_{in}$ that included the tubing for the fluidic source evidenced that driving the chip with a continuous flow made it much more sensitive to leaks. This finding led to a redesign of the chip and its setup to be able to drive it with a continuous flow for the purpose of applying pharmacokinetic drug concentration exposures [16].

### 4.4. Insights from Solving Circuit Dynamics

The circuit matrix, previously defined in Equation (7), results in a set of differential equations relating the rate of change in the volume of the *i*’th drop $Q_{ci}=dV_{i}/dt$ to the microfluidic-network design and operation parameters. Numerically solving this set of differential equations resulted in various useful values:The flow rate through each element $Q_{ij}$ was computed and could be used to estimate shear stresses within a channel [12,13]. In turn, by modifying the channel design of a microfluidic chip, shear stresses could be tuned to ensure that physiological shear stress levels are achieved in adherent cell cultures;The flow rates at nodes and their ratios could be used to estimate molecule mixing within a microfluidic network [18];Reservoir volumes $V_{ri}$ were computed and could be used to optimize the tilting scheme to ensure that flow rates remained relatively constant during the operation of a microfluidic chip [12,13];Inlet pressure $p_{in}$ for flow-rate-driven chips could be evaluated and minimized to avoid excessive pressure on fluidic sources;Drop volumes $V_{i}$ were computed as a function of time and could be used to optimize the operation of self-controlled hanging-drop setups [10];Drop volumes $V_{i}$ were computed as a function of time and could be minimized to obviate catastrophic drop failures [10,11,12].

## 5. Applications

The method outlined in this review has been applied to several open microfluidic devices designed and published within our laboratory. Previously, the modeling methodology was only described superficially, as it was out of scope of the respective papers. Here, we will give detailed information and elaborate on the insights gained by the modeling of three of these devices. The first was a standing-drop microfluidic network, developed and redesigned by Lohasz et al. [9,16,18], herein referred to as “Scalable microphysiological system”. The second was a hanging-drop microfluidic network, developed by Boos et al. [12,13], herein referred to as “Placenta-on-a-chip”. The third was a hanging-drop microfluidic network, first developed by Misun et al. [15] and redesigned by Wu Jin et al. [10], herein referred to as “Hanging-drop-based islet perifusion system”.

### 5.1. Scalable Microphysiological System

The scalable microphysiological system was the subject of the model presented in Figure 6, which was used to predict flow rates through the device that then were validated experimentally (Figure 7). This system was operated by tilting it back and forth, inducing continuous gravity-driven bidirectional flow. We cultured microtissues within the compartments that represented different organ models. The constant perfusion enabled continuous interaction and interorgan communication.

The time dependence of the flow rate could easily be determined with the model, as it was a direct output $Q_{out}$ from solving Equation (7). Experimentally, this flow rate was more tedious to validate. Setting the chip at a given tilting angle α of 10°, 20°, and 30°, a prescribed volume of 150 μL was deposited in the inlet. The liquid was subsequently completely sampled at the outlet at a given time point. The process of liquid prescription and sampling was repeated for every time point indicated by the measured values in Figure 7a–c.

With the validated model, various operational conditions could be simulated to predict device operation and optimize tilting schemes for gravity-driven flow at room (20 °C) and incubator (37 °C) temperatures. Temperature changes induced change in the medium’s viscosity and density, which impacted flow. First, the maximum achievable flow rate $Q_{in}$ at 0 min, when the prescribed volume was first added to the device, was plotted as a function of the tilting angle α (Figure 7d). Then, by computing the time before the inlet reservoir was fully emptied, the maximum tilting interval was plotted as a function of the tilting angle α (Figure 7d). Figure 7g, like Figure 7d, was also generated to predict the effect of various initial inlet reservoir liquid volumes $V_{rin}$ (100, 150, and 200 μL) on the tilting intervals. As the volumes increased, the different liquid levels increased the initial hydrostatic pressure. These tilting intervals were used to optimize chip operation to ensure continuous flow during microtissue culture and to prevent the complete draining of the elevated (source) reservoir. The optimized tilting cycle can be seen in Figure 7h and was used to generate the published results [9].

### 5.2. Placenta-on-a-Chip

The placenta-on-a-chip was the second iteration of the microfluidic multitissue platform for advanced embryotoxicity testing, invented by Boos et al. [12], that recapitulated the maternal–placental–embryonic axis [13]. The chip, shown in Figure 8a, featured two liquid phases, a “maternal” and “embryonic” side, separated by a semipermeable membrane on which placental cells grew. Placental cells were grown on the membrane in the closed maternal compartment, which was accessible by two hanging drops on either side. Embryoid bodies were cultured in hanging drops underneath the placental barrier on the embryonic side. Gravity-driven flow was induced through the network by continuously tilting the device by ±5°.

The maternal side featured a large culture chamber (Figure 8a) in which placental cells were seeded to form a confluent layer as a model of the placental barrier. We modeled the shear stresses on the placental barrier along the black cut line shown in Figure 8b. With this shear stress model, we showed that the placental barrier on the maternal–embryonic circular boundary, delineated by the dashed red lines in Figure 8b, was exposed to a uniform shear stress. We optimized the microfluidic design of the maternal side in order to minimize the shear stress by using the modeling techniques elaborated in the previous section to predict flow as a function of a tilting angle of ±5° (Figure 8c). The optimization consisted of including a resistive serpentine channel (Figure 8b) and adding additional reservoir drops to increase the volume throughput at each tilting step. We ensured that the intervals of the maximum flow rates within the maternal side were sustained as long as possible within each tilting cycle (Figure 8c) to promote cell polarization on the barrier. We achieved a maximal flow rate during chip operation of 4.6 µL min^−1^, which we used to compute the shear stress in Figure 8b.

### 5.3. Hanging-Drop-Based Islet Perifusion System

A platform to perform glucose-stimulated insulin secretion (GSIS) assays with single pancreatic islets was presented by Misun et al. [15]. The platform enabled studying the dynamics of insulin release by islet microtissues at high temporal resolution. Perifusion of single-islet microtissues produced biologically more relevant results than the typical pooling and analysis of several microtissues. The platform enabled the detection of complex pulsatile insulin secretion at single-islet resolution, while pooling of microtissues only would yield average values. The platform was adapted by Wu Jin et al. [10] to parallelize insulin sampling, and allowed for the simultaneous investigation of four pancreatic islets in separate hanging drops (Figure 9).

Like in the original device [15], sampling was carried out by applying a continuous flow through hanging drops hosting the individual islet microtissues and sampling at the outlets (green region in Figure 10A). An additional drop adjustment fluidic branch was added with a control drop to set the drop height (blue region in Figure 10A). This branch allowed automatically regulating and stabilizing drop heights throughout the network without the need of a microscopy-based feedback (in principle, setting an outlet with a needle-type valve allows keeping the network drop height constant). The self-regulation of the drop height, depicted in Figure 10B, was demonstrated experimentally [10]. The control outlet aspirated liquid, as long as the control drop ALI was lower than the needle outlet. Once the drop receded above the needle outlet, the control outlet aspirated air. This procedure repeated throughout the experiment, controlling the drop height.

However, a phenomenon known as capillary-wetting hysteresis [19,20] caused certain irregularities in the drop height control. This phenomenon, described in detail in Appendix A, induced an inhomogeneous wetting of the needle depending on whether the drop size was increasing or decreasing. The wetting hysteresis of an advancing or receding air–liquid–solid tripoint induced more complex wetting dynamics. The wetting dynamics were modeled by including a hysteresis in the drop height (dashed red line in Figure 11) and successfully recapitulated the dynamics observed experimentally. This result showed that, although the drop height of the control drop changed by ±25 μm, as the control drop withdrew liquid, the height of the hanging drops hosting islets changed substantially less by approximately ±2.5 μm.

## 6. Conclusions

This review describes a method used to model microfluidic standing- and hanging-drop networks. The technique can be used with any open microfluidic network to analyze subtle ALI dynamics and optimize chip operation with different flow sources. We show that, with basic microfluidic components and rigorous variable and circuit definitions, intricate, open microfluidic circuits can be modeled successfully. The nonlinear dynamics of an ALI can be represented by using a behavioral model, which describes the complex dependency between drop geometry, capillary and hydrostatic pressures, and flow rate with a differential equation.

The same strategy can be applied to various microfluidic chips, for example, to predict flow behavior, to optimize chip operation, to compute shear stresses on cells inside chambers and channels, to design microfluidic channels, or to model hysteresis behavior on wetted needles.

The versatility of this method has been demonstrated in several articles and conference papers [10,11,13]. Ultimately, circuit-based open-microfluidic-network design is a “first-pass” modeling technique that predicts flow. With an optimized circuit design, more intricate 3D mathematical models can be used to characterize fluid dynamics and species transport within final designs.

## Figures and Tables

**Figure 1 micromachines-13-01124-f001:**
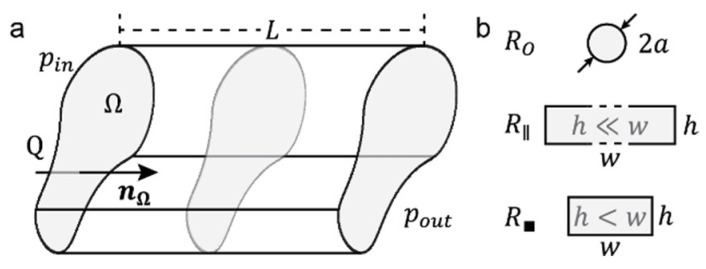
**Parameters required to define the hydraulic resistance.** (**a**) Schematic representation of a channel of constant arbitrary cross section Ω. Such channel geometry yields the most general definition of the hydraulic resistance R for a channel of length L, with a pressure difference $\Delta p=p_{in}-p_{out}$ and a normal $n_{\Omega }$ flow rate Q. (**b**) Typical cross sections of circular $R_{O}$, high-aspect-ratio rectangular $R_{∥}$ (also known as “parallel plates”), and rectangular channels $R_{◼}$. Relevant design dimensions are defined on the cross sections.

**Figure 2 micromachines-13-01124-f002:**
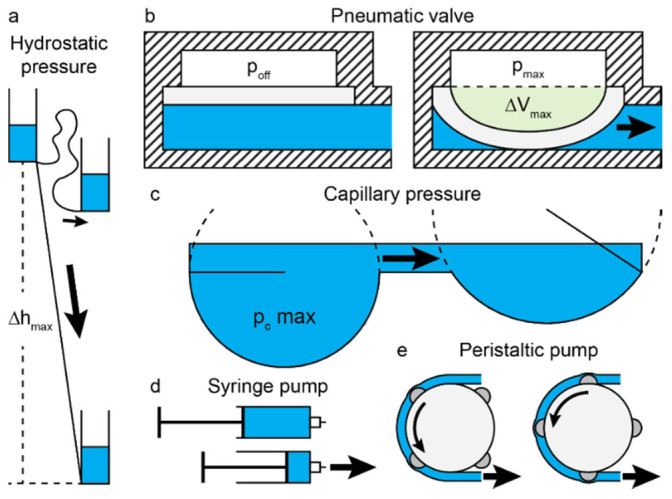
**Typical fluidic sources****.** (**a**) Hydrostatic pressure driving flow, where maximum pressure is applied when the height difference is maximal $\Delta h_{\max}$—depending on the chip design. (**b**) Pneumatic driving of flow, where maximum fluid displacement $\Delta V_{\max}$ is applied when the valve is actuated at its maximal pressure $p_{\max}$. (**c**) Capillary pressure driving flow, where maximum pressure $p_{c}\max$ is applied when the surface curvature $r_{\min}$ is minimal. (**d**) Syringe pump driving flow, where plunger leakage and syringe volume limit flow rate. (**e**) Peristaltic pump driving flow, where tubing compliance and volume limit flow rate.

**Figure 3 micromachines-13-01124-f003:**
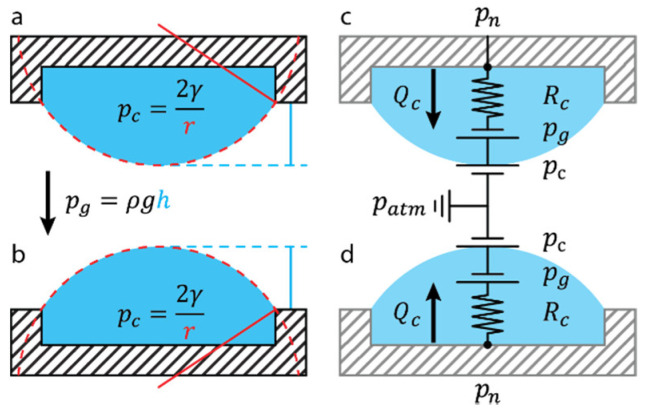
**Circuit equivalence of drop compartments.** Schematic representation of (**a**) a hanging-drop and (**b**) a standing-drop compartment and the capillary pressure $p_{c}$ applied by the curvature r of the air–liquid interface (ALI) from a reference atmospheric pressure $p_{atm}$. Circuit equivalence of fluid movement in (**c**) a hanging and (**d**) a standing drop. At equilibrium, the equivalence $p_{n}=p_{c}\pm p_{g}$ holds, and no flow is induced. An increase in $p_{n}$ due to hydrostatic pressure (e.g., by tilting the chip or elevating an inlet reservoir) or hydrodynamic pressure (e.g., fluid motion within the fluidic network) generates a net flow $Q_{c}=\Delta p/R_{c}$ toward the drop ALI. An increase in drop volume due to the ALI reduces r. A reduction in r induces an increase in $p_{c}$, which eventually reaches a new equilibrium with $p_{n}$, stopping flow toward the ALI at a new drop volume.

**Figure 4 micromachines-13-01124-f004:**
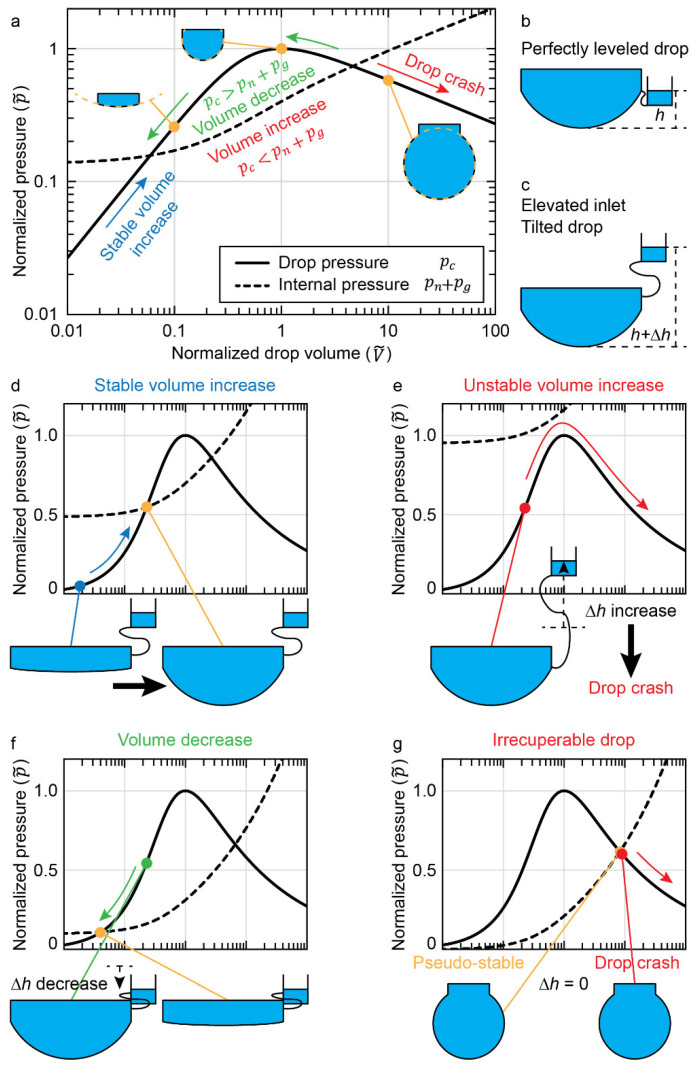
**Dynamics of a hanging-drop crash.** (**a**) Logarithmic-scale plot of the pressure, normalized by the maximum capillary pressure of a hemispherical drop ($\tilde{p}=p/\left(2\gamma /a\right)$), as a function of the volume of a hanging drop, normalized by the volume of a perfectly hemispherical drop ($\tilde{V}=V/\left(2\pi a^{3}/3\right)$). The normalized drop capillary pressure ${\tilde{p}}_{c}$ (black line) and the normalized drop internal pressure of the drop network $\left(p_{n}+p_{g}\right)/\left(2\gamma /a\right)$ (dashed black line) are plotted. ${\tilde{p}}_{c}$ is independent of the drop design and is only dependent on the normalized drop volume $\tilde{V}$. The internal pressure is given for a distorted hanging drop of 3.9 mm aperture diameter at 37 °C and subjected to a hydrostatic pressure of a 1 mm-high water column. (**b**) Schematic representation of a hanging drop, connected to a perfectly leveled reservoir. In this case, the drop would empty into the reservoir until the bottom of the ALI would be perfectly flush with the drop aperture. (**c**) Schematic representation of the typical-case scenario, where hydrostatic pressure comes from the drop height and any height difference to the microfluidic network Δh. (**d**–**g**) Plot of the normalized pressure on a linear scale with the normalized volume on a logarithmic scale. (**d**) Case of a stable drop volume increase. (**e**) Case of an unstable volume increase following an increase in internal pressure, causing the drop to crash. (**f**) Case of a volume decrease following a decrease in internal pressure. (**g**) Case of an irrecuperable drop, if the drop height cannot be reduced below the drop aperture height. A pseudostable state exists but diverges easily to a drop crash.

**Figure 5 micromachines-13-01124-f005:**
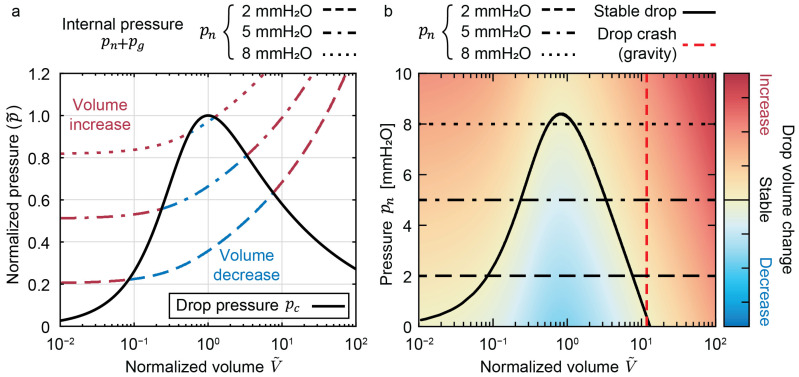
**Phase diagram of the experimental state of a hanging drop with an aperture diameter (2a) of 3 mm.** (**a**) Linear plot of pressures, normalized by the capillary pressure of a hemispherical drop $\tilde{p}$ as a function of the logarithmic drop volume, normalized by the volume of a hemispherical drop $\tilde{V}$. The normalized drop capillary pressure ($p_{c}$) from Appendix A is plotted as a solid black line. $p_{c}$ is independent of the aperture diameter. The normalized internal pressure ($p_{n}+p_{g}$) from Appendix A is plotted for a pressure $p_{n}$ equivalent to 2, 5, and 8 mm of water in various dashed lines. Subtracting $p_{c}$ by $p_{n}+p_{g}$ indicates whether liquid is driven into (volume increase in red) or out of (volume decrease in blue) the drop. (**b**) Phase diagram of drop volume change with applied pressures $p_{n}$ in mm of water as a function of normalized drop volumes $\tilde{V}$. The three pressures plotted in subfigure (**a**) are plotted as an indicator. The normalized subtraction $p_{c}-p_{n}-p_{g}$ is plotted on the z axis as a color map. Increasing the pressure, via a flow rate generating hydrodynamic pressure or a column of water generating hydrostatic pressure, causes a vertical translation upward in the phase diagram. A volume decrease (when in the blue area of the phase diagram) causes a horizontal translation leftward. A volume increase (when in the red area of the phase diagram) causes a horizontal translation rightward. The drop volume stops changing when it reaches the “Stable drop” line in black. A volume increase past the dashed red line incurs a drop crash due to gravitational forces.

**Figure 6 micromachines-13-01124-f006:**
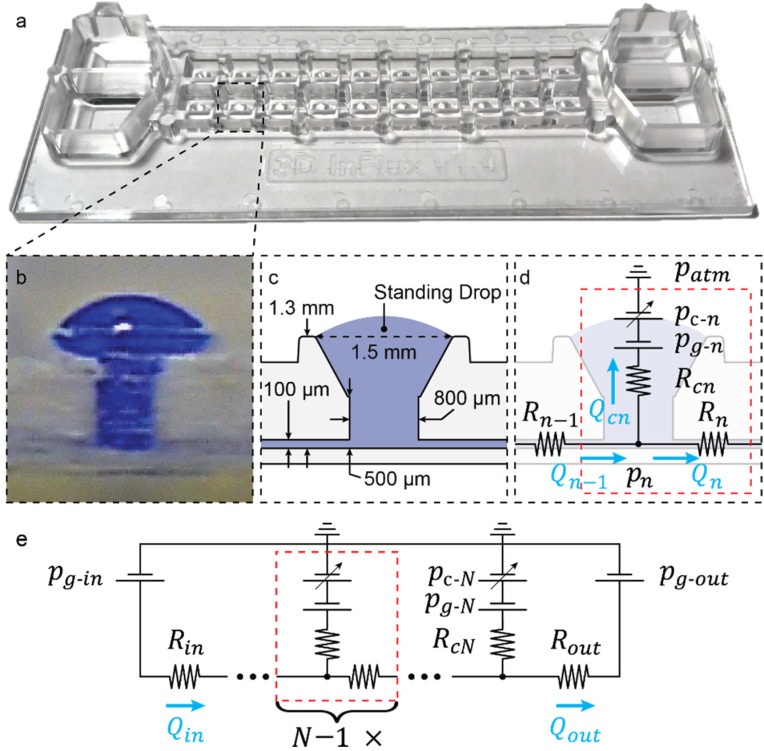
**Circuit-based-microfluidic-network design example for a standing-drop chip configuration.** (**a**) Photograph of the previously published “scalable microphysiological system” [9]. Dashed black frames represent schemes of a standing-drop compartment of the chip. (**b**) Photo of the cross section of a single compartment filled with blue-dyed water. (**c**) Schematic of a standing-drop compartment with relevant dimensions for modeling resistances and pressures. (**d**) Electrical-equivalent-circuit representation of a single standing drop. All known variables (resistances R, hydrostatic pressures $p_{g}$, and capillary pressures $p_{c}$) are defined in the scheme. All unknown variables (hydrodynamic pressures $p_{i}$ and flow rates $Q_{i}$) are defined, and the flow directions and rates are set. A reversal of the flow direction is indicated by a negative flow rate. The dashed red frame outlines a single standing drop unit to be modeled and repeated. (**e**) Equivalent circuit of the entire “scalable microfluidic system”. The single standing drop (dashed red frame) is repeated N = 10 times for the entire chip. Known ($R_{in/out}$ and $p_{g-in/out}$) and unknown ($Q_{in/out}$) inlet and outlet variables are included to show the complete circuit.

**Figure 7 micromachines-13-01124-f007:**
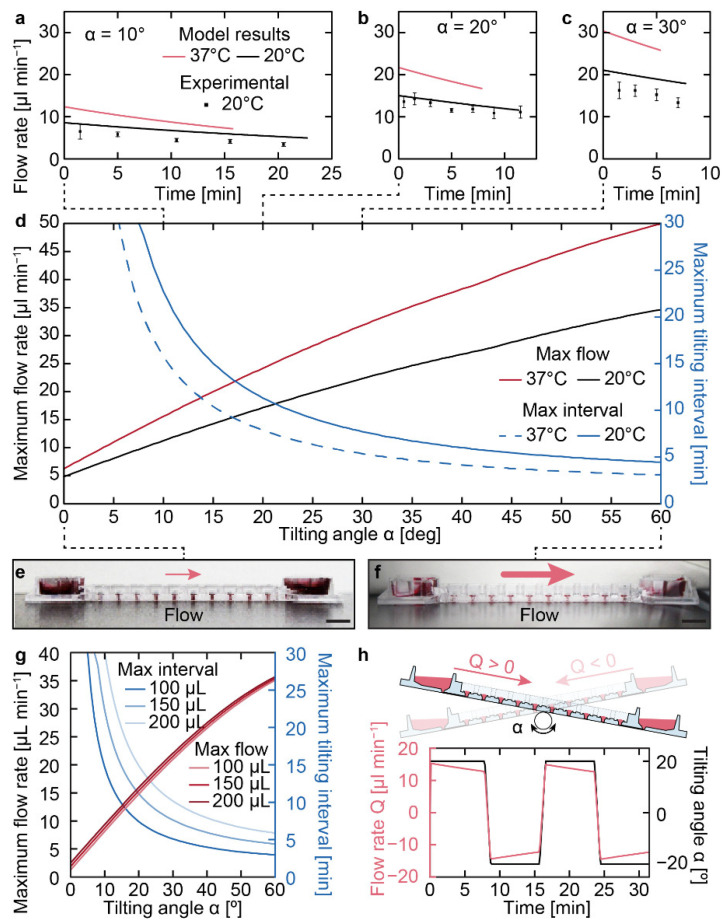
**Tilting-angle-dependent perfusion of the chip using an initial liquid volume of 150 μL.** (**a**–**c**) Flow rate as a function of time at tilting angles α of (**a**) 10°, (**b**) 20°, and (**c**) 30°. Measured values at 20 °C (12 measurements; data represented as mean ± SD) matched the calculated flow rates over time and allowed for extrapolation to an experimentally relevant temperature of 37 °C. Slopes of the decreasing flow rates over time at 20 °C amounted to −0.16 ± 0.018, −0.30 ± 0.037, and −0.54 ± 0.104 μL min^−2^ for the measured values and −0.16 ± 0.001, −0.30 ± 0.001, and −0.41 ± 0.001 μL min^−2^ for the calculated values at tilting angles of 10°, 20°, and 30°, respectively. (**d**) Maximum flow rate and tilting interval until drainage of the top reservoir at 20 and 37 °C as a function of the tilting angle. The curves for initial liquid volumes of 100 and 200 μL can be found in subfigure (**g**). Representative side views of the chip at (**e**) 0° and (**f**) 60° showing the stability of the standing drops at large tilting angles (scale bars = 5 mm). (**g**) The maximum flow rate and tilting interval until drainage of the top reservoir at 37 °C as a function of the tilting angle was calculated for initial liquid volumes of 100, 150, and 200 μL. Lower liquid volumes resulted in lower flow rates and shorter applicable tilting intervals. (**h**) Bidirectional flow rate through the channel upon repeated tilting over two tilting cycles with the following parameters: tilting angle α = 20°, tilting interval = 7 min, and transition time = 50 s. Adapted with permission from Lohasz et al. [9]. 2018, Elsevier.

**Figure 8 micromachines-13-01124-f008:**
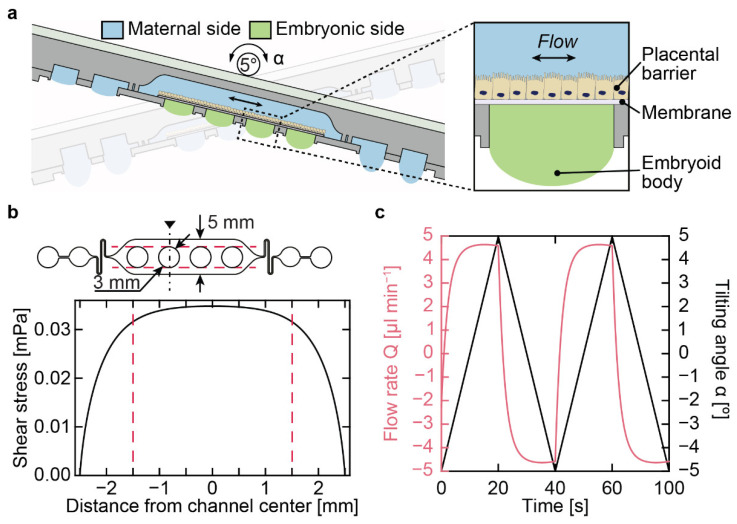
**Optimized design and modeling results of the placenta-on-a-chip.** (**a**) Side view of the placenta-on-a-chip device, showing the maternal and embryonic sides, separated by the placental barrier, grown on a permeable membrane. An embryoid body was used as the biological model on the embryonic side. Tilting the chip by ±5° induced flow in the closed maternal compartment. (**b**) The hydraulic resistance of the serpentine channel and the timing of the tilting protocol were optimized to minimize fluid shear stresses on the placental barrier while keeping it as constant as possible to promote cell polarization on the barrier. The aperture of the hanging drops is indicated by a red dotted line and constituted the direct interface between the maternal and embryonic side. A constant shear stress of 0.034 mPa was obtained across the hanging-drop aperture with the optimized flow rate. (**c**) Flow rates (pink) in relation to the applied tilting angle (black) over time. The tilting angle of ±5° and transition time of 20 s represent optimized parameters to obtain uniform and maintained flow rates with short transition times. The maximum flow rate in the maternal cell culture channel was 4.6 µL min^−1^. Adapted with permission from Boos et al. [13] (CC BY-NC-ND 4.0). 2020, John Wiley & Sons—Books.

**Figure 9 micromachines-13-01124-f009:**
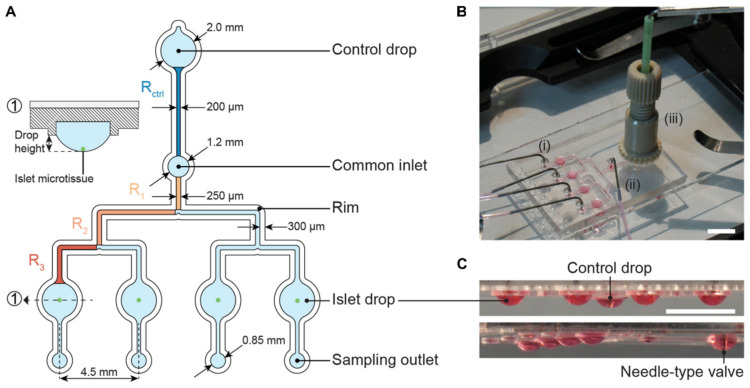
Microfluidic hanging-drop perifusion system. (**A**) Chip layout and dimensions. The fluidic structures (light blue) have a depth of 500 µm, except for the common inlet that has a recess depth of 1 mm. The hydrophobic rim structures (white) defined the fluidic channels and hanging drops. They had a height of 250 µm measured from the chip surface. Highlighted channel sections in dark blue and orange were considered for the calculation of the hydraulic resistance in the channels. The channels between common inlet and control drop and between common inlet and islet drops were designed to have the same hydraulic resistance. (1) Cross-sectional view of the islet drop with the islet microtissue at the bottom of the hanging drop. (**B**) Top view of the assembled chip with (i) four parallel outlet tubes, (ii) one common inlet tube, and (iii) a NanoPort assembly with a needle-type valve inserted at the center. Scale bar: 10 mm. (**C**) Side views of the chip with hanging drops visualized with red dye. All hanging drops had equal sizes and shapes. Scale bar: 5 mm. Reproduced from Wu Jin et al. [10] (CC BY 4.0).

**Figure 10 micromachines-13-01124-f010:**
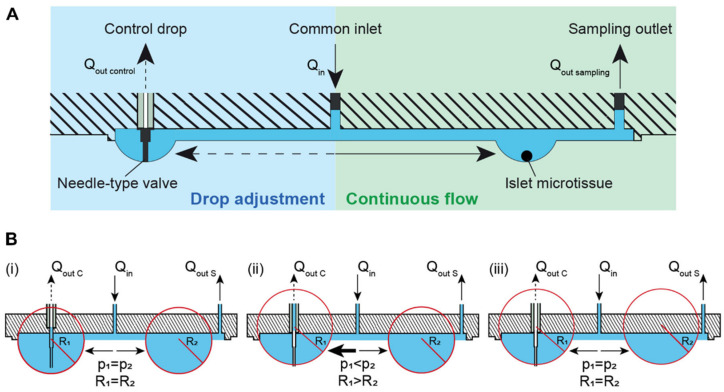
Theory and concept of the automatic hanging-drop-size adjustment in the microfluidic chip. (**A**) The inflow ($Q_{in}$) was split into two flows, one toward the islet drops (right) and the other toward the control drop (left). There was a continuous flow of liquid from the inlet toward the islet drops due to active sampling from the sampling outlets at the rate of $Q_{\mathrm{out}\;\mathrm{sampling}}$. An irregular flow was observed from the inlet toward the control drop due to the alternating withdrawal of liquid and air from the needle-type valve at a rate of $Q_{\mathrm{out}\;\mathrm{control}}$. (**B**) Self-regulation of drop heights between interconnected hanging drops. Liquid was constantly added into the system at a rate of $Q_{in}$ and withdrawn through the sampling outlet at a rate of $Q_{\mathrm{out}\;S}$ and through a needle-type valve at a rate of $Q_{\mathrm{out}\;C}$. (i) Two hanging drops in equilibrium with identical Laplace radius and pressure. The inflow was evenly distributed to the two drops. As long as the tip of the needle-type valve was immersed in the drop, liquid removal through the valve occurred. Consequently, (ii) the size and the Laplace pressure of the two drops became different with drop radius and pressure being inversely correlated. Due to the pressure difference, there was an increased flow toward the control drop (left) with the lower pressure. As soon as the tip of the needle-type valve was exposed to air, only air was withdrawn. The control drop remained stable at the height defined by the valve needle length. (iii) In the next step, the two drops reached equal Laplace radius and pressure through equilibration through the liquid phase. The system was now stable, and the drop height in control and tissue drop was maintained constant. Reproduced from Wu Jin et al. [10] (CC BY 4.0).

**Figure 11 micromachines-13-01124-f011:**
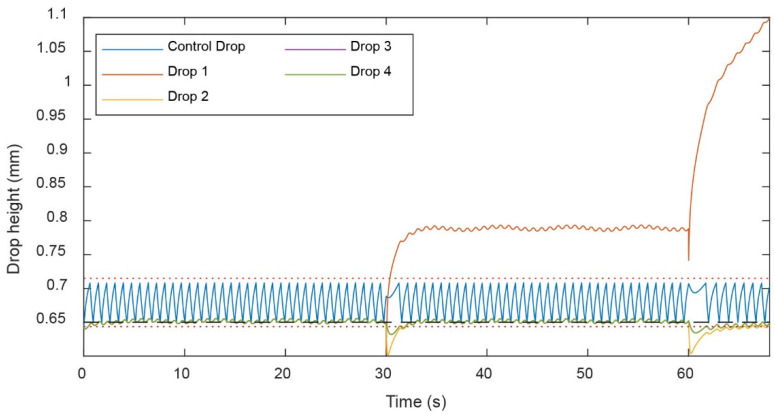
Matlab simulation showing the effect of wetting of the rim. The control drop height was set to 650 μm, indicated as black dashed line. Red dotted lines indicate the maximum and minimum drop heights of the control drop, as explained through the drop hysteresis in Figure 10B. The blue oscillatory pattern indicates the constant readjustment of the control drop height due to the pulsed withdrawal of excess medium through the needle-type outlet. The inflow rate was 80 μL/min, outflow rate was 15 μL/min for each drop, and the control needle outflow rate was 45 μL/min. A spontaneous wetting instance was simulated at 30 s and 60 s, where the aperture of drop 1 increased by 5% (10 μm) each time.

**Table 1 micromachines-13-01124-t001:** List of common fluidic sources, driving forces, defining equations, and physical limitations.

Name	Driving Force	Defining Equation	Physical Limitation	
			*Pressure Limitation*	*Volumetric Limitation*
Capillary pressure	Surface tension	$p_{c}=\frac{2\gamma }{r}$	Minimum surface curvature	Drop volume
Hydrostatic pressure	Gravity	$p_{g}=\rho gh$	Maximum height difference	Reservoir volume
Pneumatic valve	Pneumatic	$Q\propto \Delta V\propto p_{\mathrm{air}}$	Air pressure	Valve volume
Syringe pump	Mechanical	$Q\equiv Q_{\mathrm{set}}$	Plunger leakage	Syringe volume
Peristaltic pump			Tubing compliance	Tubing volume

**Table 2 micromachines-13-01124-t002:** Useful relationships between the various geometric elements of a spherical cap are included herein. Equations for subhemispherical caps are written in blue and equations for suprahemispherical caps are written in green.

Geometric Element	f(r,h)	f(r,a)	f(a,h)	f(V,a)
**Volume** V	$\frac{\pi }{3}h^{2}\left(3r-h\right)$	$\frac{\pi }{3}\left(2r^{3}-\left(2r^{2}+a^{2}\right)\sqrt{r^{2}-a^{2}}\right)$ $\frac{\pi }{3}\left(2r^{3}+\left(2r^{2}+a^{2}\right)\sqrt{r^{2}-a^{2}}\right)$	$\frac{\pi }{6}h\left(3a^{2}+h^{2}\right)$	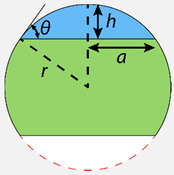
**Cross-sectional area** A	$r^{2}\mathrm{asin}\left(\frac{\sqrt{2rh-h^{2}}}{r}\right)+\sqrt{2rh-h^{2}}\left(h-r\right)$	$r^{2}\mathrm{asin}\left(\frac{a}{r}\right)-a\sqrt{r^{2}-a^{2}}$ $r^{2}\mathrm{asin}\left(\frac{a}{r}\right)+a\sqrt{r^{2}-a^{2}}$	${\left(\frac{h^{2}+a^{2}}{2h}\right)}^{2}\mathrm{asin}\left(\frac{2ah}{h^{2}+a^{2}}\right)+a\frac{h^{2}-a^{2}}{2h}$	
**Surface curvature** r	r=r	r=r	$\frac{h^{2}+a^{2}}{2h}$	$\frac{1}{12V}\left(\pi a^{4}+\beta +\frac{\pi^{2}a^{8}}{\beta }\right),$ $\beta =\frac{{\left(3V+\sqrt{\pi^{2}a^{6}+9V^{2}}\right)}^{\frac{4}{3}}}{\pi^{\frac{1}{3}}}$
**Cap height** h	h=h	$r-\sqrt{r^{2}-a^{2}}$ $r+\sqrt{r^{2}-a^{2}}$	h=h	$\frac{{\left(3V+\sqrt{\pi^{2}a^{6}+9V^{2}}\right)}^{\frac{2}{3}}-\pi^{\frac{2}{3}}a^{2}}{\pi^{\frac{1}{3}}{\left(3V+\sqrt{\pi^{2}a^{6}+9V^{2}}\right)}^{\frac{1}{3}}}$
**Cap width** a	$\sqrt{2rh-h^{2}}$	a=a	a=a	a=a
**Contact angle** θ	$\mathrm{asin}\left(\frac{\sqrt{2rh-h^{2}}}{r}\right)$	$\mathrm{asin}\left(\frac{a}{r}\right)$	$\mathrm{asin}\left(\frac{2ah}{h^{2}+a^{2}}\right)$	Not useful

**Table 3 micromachines-13-01124-t003:** Variable definition for each microfluidic element. Subscripts in the nomenclature are descriptive of the element, where c, *r*, *v*, and *g* denote capillary, reservoir, valve, and gravity. Subscripts are also enumerative, where i denotes the *i*’th node, ij denotes an element between the *i*’th and *j*’th nodes, and *in* and *out* denote the inlet and outlet.

Microfluidic Element	Example	Nomenclature	Variable Properties	
Hydraulic Resistance	Filled channel	$R_{ij}\;$	*Known*	*Constant*
	Drop volume	$R_{ci}\;$		*Variable*
	Reservoir in/outlet	$R_{ri}\;$		
	Valve	$R_{vi}\;$		
Flow rate	Syringe-driven in- or outflow	$Q_{in},\;Q_{out}\;$		*Constant*
		$Q_{in}\left(t\right),\;Q_{out}\left(t\right)\;$		*Variable*
Capillary pressure	Drop ALI	$p_{ci}\;$		*Variable*
Hydrostatic pressure	Chip level	$p_{gi}\;$		*Constant*
	Chip tilting	$p_{gi}\left(\theta \right)\;$		*Variable*
Flow rate	Filled channel	$Q_{ij}\;$	*Unknown*	*Computed numerically*
	Drop volume	$Q_{ci}$		
	Reservoir	$Q_{ri}$		
	Valve	$Q_{vi}$		
Pressure	Nodal pressure	$p_{i}$		

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
