# Peer review of "Circuit-Based Design of Microfluidic Drop Networks"

_micromachines, 2022, doi:10.3390/mi13071124_

Round 1

Reviewer 1 Report

The manuscript is of good quality, well-written and nicely organized, in my opinion. But the figure captions are too long and also a graphical abstract would attract more readers. 

Reviewer 2 Report

The authors present an interesting study involving a method to model hanging and standing drop networks using the hydraulic-circuit analogy. The methods are very detailed, well written and transparent resulting in a smooth read. The applications are well presented and informative, however we have the following comments about this manuscript that should be addressed prior to publication:

Comment 1: This paper seems to describe the methods used in several already published papers. This begs the question: why was this methodology not fully described in past publications? In other words what is the purpose of this paper if other papers have already successfully used the presented methods and described them? Additionally, the authors should either present or cite publications where the ALI method is tested on academic examples.

Comment 2: Table 3 should be re-formatted such that it fits on a portrait page. As it is, it is not easily readable without straining one’s neck. There are also some display issues with the square roots and parentheses (Figure 1)

Reviewer 3 Report

Thank you for inviting me to review this manuscript titled Circuit-based design of microfluidic drop networks. In this paper, the authors reported a circuit-based model for open microfluidic networks. Using this first pass method, microfluidic standing and hanging drop networks were successfully modeled, showing its feasibility in charactering fluid dynamics, chip designing and operating. This quality work is scientifically interesting and shows practical significance in open microfluidic applications. Therefore, I recommend this paper to be published on Micromachines. The only thing I need to mention is that, the numbering of equation was inconsistent with that in the text, please check.
